# Dynamic hyperinflation induced by the 6-minute pegboard and ring test in hospitalized patients with exacerbated COPD

**Cassia Fabiane de Barros** [1], **Rosimeire Marcos Felisberto** [1], **Kelly Cristina Albanezi Nucci** [1], **Andre Luis Pereira de Albuquerque** [1], **Elaine Paulin** [2], **Christina May Moran de Brito** [1], **Wellington Pereira Yamaguti** [1] *

**1** Hospital Sírio-Libanês, São Paulo, SP, Brasil, **2** Universidade do Estado de Santa Catarina (UDESC), Florianópolis, SC, Brasil

* wellington.psyamaguti@hsl.org.br, wellpsy@yahoo.com.br

**Data Availability Statement:** All relevant data are within the paper.

## Abstract

### Background

The six-minute pegboard and ring test (6-PBRT) has been used to evaluate functional capacity of the upper limbs in stable chronic obstructive pulmonary disease (COPD) patients. To the best of our knowledge, no studies have evaluated dynamic hyperinflation (DH) during exercise with upper limbs in the hospital setting. The aim of this study was to evaluate physiological responses and DH induced by 6-PBRT in hospitalized patients with acute exacerbation of COPD (AECOPD).

### Methods

A cross-sectional study was conducted in a tertiary hospital enrolling patients who were hospitalized due to AECOPD. All included participants underwent an evaluation of lung function and 6-PBRT when they reached minimum clinical criteria. Ventilatory and hemodynamics parameters were monitored during 6-PBRT and until 6 minutes of rest after the test. Symptoms of dyspnea and upper limb fatigue were also measured.

### Results

Eighteen patients (71.3±5.1 years) with a mean $FEV_1$ of 43.2±18.3% were included in the study (11 females). Prevalence of DH after 6-PBRT was 50% (considering the drop of 150 ml or 10% of inspiratory capacity, immediately after the end of the test). There was a significant increase in respiratory rate, minute volume, dyspnea, and upper limb fatigue after the end of 6-PBRT ($p<0.05$). Dyspnea recovered more precociously than the perception of fatigue, being reestablished within four minutes of rest. An increase in heart rate, systolic and diastolic blood pressures was also induced by 6-PBRT ($p<0.05$), requiring 6 minutes of recovery to return to baseline. No adverse events were observed during 6-PBRT. We concluded that 6-PBRT induces physiological changes during its execution, at safe levels, requiring a maximum of 6 minutes for recovery. Finally, the test proved to be safe and applicable for patients hospitalized due to AECOPD.

**Funding:** Brazilian Scientific Agency Fundação de Amparo à Pesquisa do Estado de São Paulo (FAPESP, number 18/16832-9).

**Competing interests:** The authors have declared that no competing interests exist.

## Introduction

Chronic obstructive pulmonary disease (COPD) is an important cause of morbidity and mortality in the world and represents a large proportion of users of the health system [1]. The natural course of the disease is punctuated by aggravation episodes, known as exacerbations, leading to deterioration and mortality associated with the disease [2, 3]. Acute exacerbation of COPD (AECOPD) contributes to worsening lung function and symptoms [3], reduced activities of daily living (ADL) [4], altered muscle function [5], a substantial decline in functional status [4, 6] and quality of life [1], and increased morbidity and mortality [1, 5].

Ventilatory impairment continues to be the main factor limiting exercise capacity for most patients. Increased ventilator demand during physical activity may hyperinflate patients with COPD due increased functional residual capacity (FRC) and decreased inspiratory capacity (IC) [7]. Exercise intolerance involving upper limbs is due to increased ventilator and metabolic demand, development of dynamic hyperinflation, and thoracoabdominal asynchrony during these activities [8–10]. It is assumed that such changes are responsible for the greater sensation of fatigue and dyspnea leading to early interruption during activities, simple ADLs of self-care, such as eating, personal hygiene, bathing, washing hair, and getting dressed [11], especially those performed without support [12].

Some tests have been described in the literature and are recognized as simple, valid and reproducible for assessing the functional capacity of upper limbs in stable COPD, such as the Unsupported Upper Limb Exercise Test (UULET) [13], Grocery Shelving Task (GST) [14] and 6-minute pegboard and ring test (6-PBRT) [15]. The 6-PBRT is the most commonly used and was first described by Celli et al. [16]. The relationship between 6-PBRT and parameters of pulmonary function [15, 17], strength and endurance of upper limbs [18, 19], incremental upper limb test [18], and upper-extremity physical ADLs [17] has been demonstrated in patients with stable COPD. In addition, 6-PBRT has also been used to show the efficacy of upper limb exercise training programs [20] since its responsiveness has already been documented [18, 21].

However, these authors did not assess inspiratory capacity reduction and the development of dynamic hyperinflation following 6-PBRT. Recent studies have observed the development of dynamic hyperinflation (DH) after an increase in load during exercise with cycle ergometer for upper limbs in patients with COPD [22] and the direct influence of exercise modality with upper limbs on the occurrence of dynamic hyperinflation [8]. Despite this safety and applicability, to date, there are no studies that have evaluated DH during 6-PBRT in hospitalized patients with AECOPD.

Considering that ventilatory limitation, characterized by dynamic hyperinflation, is an important cause for the interruption of physical activity, it is necessary to know the tests to evaluate the functional capacity at the time of AECOPD, as well as their applicability and safety in the hospital setting, and thus in the future guide pulmonary rehabilitation programs focusing on upper limb training in the hospital phase. Therefore, this study aims to evaluate ventilatory and hemodynamic responses; to verify the prevalence of dynamic hyperinflation during the 6-minute pegboard and ring test (6-PBRT); and also to evaluate the safety and applicability for hospitalized patients for an acute exacerbation of COPD.

## Materials and methods

### Ethics and participants

We conducted a cross-section study on hospitalized patients with exacerbated COPD at Hospital Sírio-Libanês from July 1, 2015 to July 30, 2016. The sample was obtained consecutively,

and patients of both genders were recruited, admitted at the hospital for treatment of AECOPD. Inclusion criteria were as follows: 1) patients with previous diagnosis of COPD before hospitalization [1]; 2) patients with exacerbated COPD classified as level II according to the American Thoracic Society (ATS) and the European Respiratory Society (ERS) [23]; 3) without cognitive or motor deficit observed from the initial evaluation which limited the performance of the tests; 4) absence of heart or other pulmonary disease previously diagnosed by the patient's physician; 5) patients who had not undergone recent thoracoabdominal surgery within one month; 6) body mass index <35 kg/ m$^2$; 7) age <85 years; and 8) without use of vasoactive drugs. The following exclusion criteria were considered: 1) inability to perform the evaluations within the criteria of technical acceptability according to methodological description and 2) cardiorespiratory instability during the tests (intense dyspnea, arrhythmias, angina, elevated heart rate above 80% of maximal heart rate predicted by age and peripheral oxygen saturation below 88% refractory to oxygen supplementation). All participants included in the study signed the Informed Consent Term, previously approved by the Ethics Committee of the Hospital Sírio-Libanês (number HSL2014-66).

## Study design and experimental procedures

Patients in the present study underwent to an evaluation protocol performed on a single day. The protocol was applied as soon as the patients presented the following minimum clinical criteria: use of noninvasive mechanical ventilation for less than 2 hours per period of 6 hours, resting dyspnea less than 7 (very intense) on the modified Borg scale, respiratory rate less than 25 incursions per minute, SpO$_2$ greater than 88% (considering the use of oxygen supplementation) and absence of a paradoxical respiratory pattern. All the patients performed the pulmonary function test and 6-PBRT. Two specific questionnaires related to lung disease were applied to the patients: one of the influence of dyspnea and fatigue in the ADL—the Pulmonary Functional Status Dyspnea Questionnaire (PFSDQ-M). This questionnaire is composed of three domains: dyspnea influence on ADLs, influence of fatigue on ADLs, and change in ADLs in comparison to the period before the disease onset [24]. Upper-extremity ADLs were assessed using PFSDQ-M specific to patients with COPD and validated for the Brazilian population [25]; and another on the impact of COPD symptoms—the COPD Assessment Test (CAT), using a validated version for the Brazilian population [26]. Anthropometric data, personal antecedents and life habits of each participant were also collected. Medical treatment was carried out according to local guidelines and included steroids, antibiotics, oxygen, and bronchodilators. Individual doses of each were titrated, and the decisions about admittance and discharge were made by the physician, who was not involved with the study protocol.

We requested that the short-term bronchodilator should not be performed 2 hours before the tests ensuring that similar conditions were established for all patients during the tests. Pulmonary function test was performed according to international guidelines using a portable spirometer (Koko pulmonary function testing model; nSpire Health Company, Longmont, CO, USA) previously calibrated. At least three acceptable maneuvers and two repeatable maneuvers were performed. The highest values obtained for each of the spirometric variables were considered, which were expressed in absolute and in percentage of the expected values of normality according to the methods and criteria recommended by the ATS and ERS. [27]. The 6-PBRT was used for assessment of functional capacity of upper limbs. The test consisted of moving the rings from the on one level to the other two pins fixed on a higher level on a vertical support. Two pins were positioned at shoulder height and the other two 20 cm above the shoulder level. A total of 10 rings (weighing 50 g each) were placed on each lower pin.

Participants were instructed to use both hands simultaneously and move the rings from the lower level to the upper level. After placing all the rings on the upper level, participants were asked to move the rings to the lower level and so on. This cycle should be repeated as often as possible in 6 minutes. The final score was the total number of rings moved. The test could be stopped if there was intense dyspnea, fatigue or any other serious discomfort, and participants were encouraged to retake the test as soon as possible, without the timer being stopped. The evaluator encouraged participants to use standardized phrases every minute during the test, following the ATS recommendations [28, 29]. Briefly, individuals were instructed to use both hands simultaneously and move the rings from the lower level to the upper one on a vertical support. After positioning all the rings on the upper level, the subjects repositioned the rings on the lower level and so on. The individual was instructed to repeat the cycle as many times as possible in 6 minutes, the final score being the total number of rings displaced. Ventilatory and hemodynamics parameters were monitored during 6-PBRT and until six minutes of rest after the test. Dyspnea and fatigue of upper limbs were also measured using the Modified Borg Scale.

## Dynamic hyperinflation assessment

Dynamic Hyperinflation (DH) was evaluated by means of serial inspiratory capacity (IC) measurements by the slow vital capacity maneuver. The maneuvers were performed at rest, immediately after the test and every two minutes after the end of the 6-PBRT, up to a total time of six minutes. Minute volume was evaluated before and after the 6-PBRT. Assessments were performed using a hand-held portable spirometer (Spiropalm 6MWT; Cosmed, Rome, AL, Italy), with a silicon face mask coupled to the patient by means of a headcap and an elastic belt. DH was defined as a decrease in IC at the end of the 6-PBRT, of at least 150 ml or a decrease of 10%, compared to the IC obtained at rest (**Fig 1**) [22, 30].

## Feasibility and safety

In order to evaluate the feasibility of the test, we verified the time of assembly and displacement of the device to the destination unit where the patient was hospitalized and the frequency of patients who were able to perform the test according to their standardization.

Regarding safety, the following adverse events were considered: desaturation during exercise (SpO$_2$ <85), loss of access (central or peripheral), loss of nasoenteral tube, dizziness, headache, hypotension (MAP <65 mmHG), angina, tachycardia (HR >80% of maximal HR),

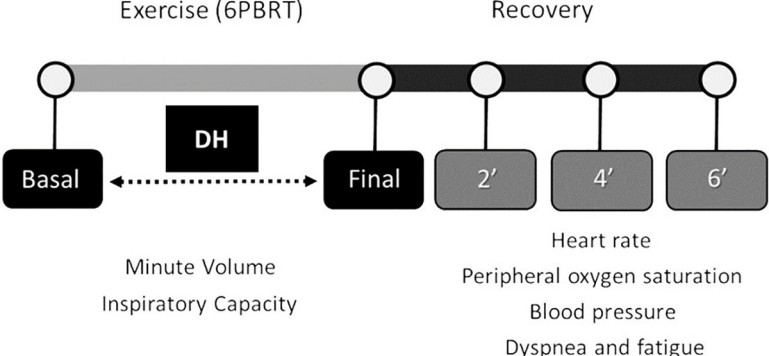

**Fig 1. Symptoms, ventilator and hemodynamics measurements during and after 6-PBRT. Abbreviations:** 6-PBRT, six-minute pegboard and ring test; DH, dynamic hyperinflation.

bradycardia (HR <50 bpm), muscular or limiting joint pain, severe dyspnea (modified Borg scale > 8), hypertension (MAP >120 mmHg) and intense sweating.

## Statistical analyses

A statistical package (SigmaPlot 11.0; Systat Software Inc.) was used for the statistical analysis of the data. Data distribution was analyzed by the Shapiro-Wilk test. According to normality in data distribution, data were described as mean and standard deviation or median and interquartile range. Comparison of IC, modified Borg scale, respiratory rate, $SpO_2$, HR, systolic blood pressure and diastolic blood pressure measured before, immediately after, and every two minutes after the 6-PBRT was performed by the simple variance analysis (repeated measures ANOVA) test. The comparison of variables between the group of patients who presented DH and did not present DH during 6-PBRT was performed using Student's t test (parametric data) and Wilcoxon (non-parametric data). For all tests, p <0.05 was considered statistically significant. Based on the results of the study by Colucci et al. (2010), who found a difference in inspiratory capacity values obtained before and after exercise of upper limbs of 290 ± 220 ml and estimating to observe a similar effect, using a 5% error (test power = 80%), it was necessary to include 17 patients in the study.

## Results

A total of 73 hospitalized patients with exacerbated COPD were screened: 41 patients did not meet the inclusion criteria, 12 patients refused, and 2 patients dropped out of the study. The final sample consisted of 18 individuals (Fig 2).

The performance in 6-PBRT was worse in patients with AECOPD (244.88 ± 63.20 number of rings moved) when compared to the average of expected reference values for normal population (375.44 ± 21.63 predicted number of rings moved) (p < 0.001) [31]. Patients who hyperinflated in the test performed worse when compared to those who did not

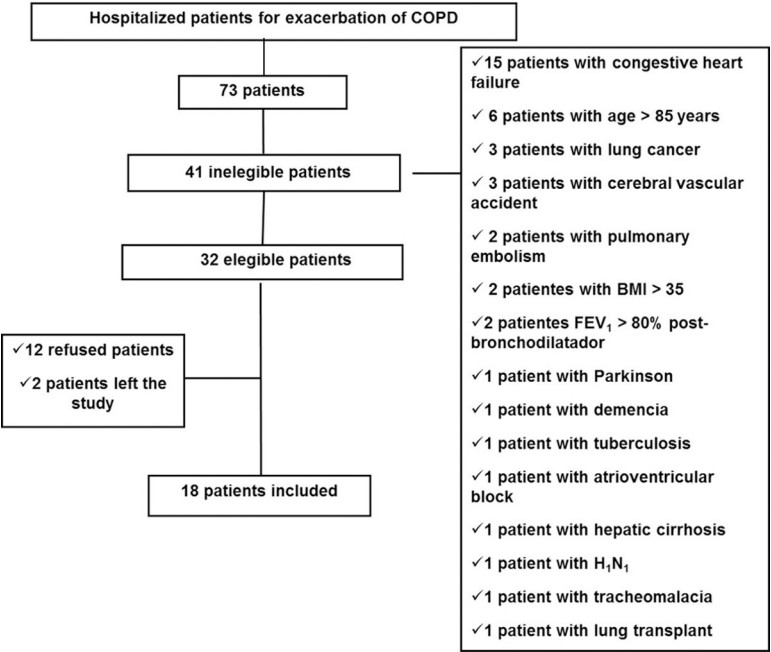

**Fig 2. Screening flow chart. Abbreviations:** AECOPD, acute exacerbation of chronic obstructive pulmonary disease.

**Table 1. Demographic and spirometric characteristics of the study participants.** Data are presented by mean ± standard deviation.

| Variables | Patients with AECOPD (n = 18) |
|---|---|
| Age (years) | 71.28 ± 5.14 |
| Gender (F/M) | 11/7 |
| Body Mass (Kg) | 70.05 ± 13.40 |
| Height (m) | 1.65 ± 0.09 |
| BMI ($Kg/m^2$) | 25.90 ± 5.07 |
| CAT | 26.38 ± 9.34 |
| PFSDQ-M fatigue | 44.88 ± 25.74 |
| PFSDQ-M dyspnea | 43.16 ± 27.45 |
| PFSDQ-M change in ADLs | 49.38 ± 28.44 |
| Oxygen supplementation at rest | 5 |
| Oxygen supplementation during test | 9 |
| GOLD classification (II / III / IV) | 7/5/6 |
| **Lung funtion post-bronchodilator** | |
| $FEV_1$ (% predicted) | 43.22 ± 18.33 |
| FVC (% predicted) | 62.61 ± 19.91 |
| $FEV_1$/FVC | 0.50 ± 0.12 |
| VC (% predicted) | 61.22 ± 19.53 |
| IC (L) | 1.51 ± 0.45 |

**Abbreviations:** AECOPD, acute exacerbation of chronic obstructive pulmonary disease; n, number of individuals; F, female; M, male; kg, kilograms; m, meters; BMI, body mass index; $m^2$, square meters; CAT, COPD Assessment Test; PFSDQ-M, Pulmonary Functional Status Dyspnea Questionnaire; ADLs, activities of daily living; GOLD, Global Initiative for Chronic Obstructive Lung Disease; $FEV_1$, forced expiratory volume in the first second; FVC, forced vital capacity; VC, vital capacity; IC, inspiratory capacity; L, liters.

hyperinflate (230.22 ± 68.56; 259.55 ± 57.49 respectively), however this result was not significant (P = 0.34).

The demographic and spirometric characteristics of the 18 patients are shown in **Table 1.**

## Ventilatory responses during and after 6-PBRT

There was a reduction in IC in the post-test evaluation but with no significant difference (p>0.05). However, when we analyzed the patients individually, we observed that nine patients (50%) presented DH during 6-PBRT (reduction of 150ml or 10% in the inspiratory capacity). We also observed that after six minutes at the recovery phase, only three patients (16.6%) still fulfilled the criteria for DH (**Fig 3**).

Patients presented an increase in respiratory rate after the test (p = 0.04), recovering completely with/within 2 minutes of rest. Regarding the minute volume, we observed a statistically significant change (p = 0.02) when comparing basal values and the values immediately after the end of the test (**Fig 4**). There was no difference in peripheral oxygen saturation during and after 6-PBRT (p>0.05).

## Hemodynamic responses during and after 6-PBRT

We observed a significant increase in HR immediately after the test (p<0.0001). For systolic blood pressure (SBP), we observed statistically significant differences immediately after the end of 6-PBRT (p<0.0001) and in the second minute of recovery after the test (p = 0.02).

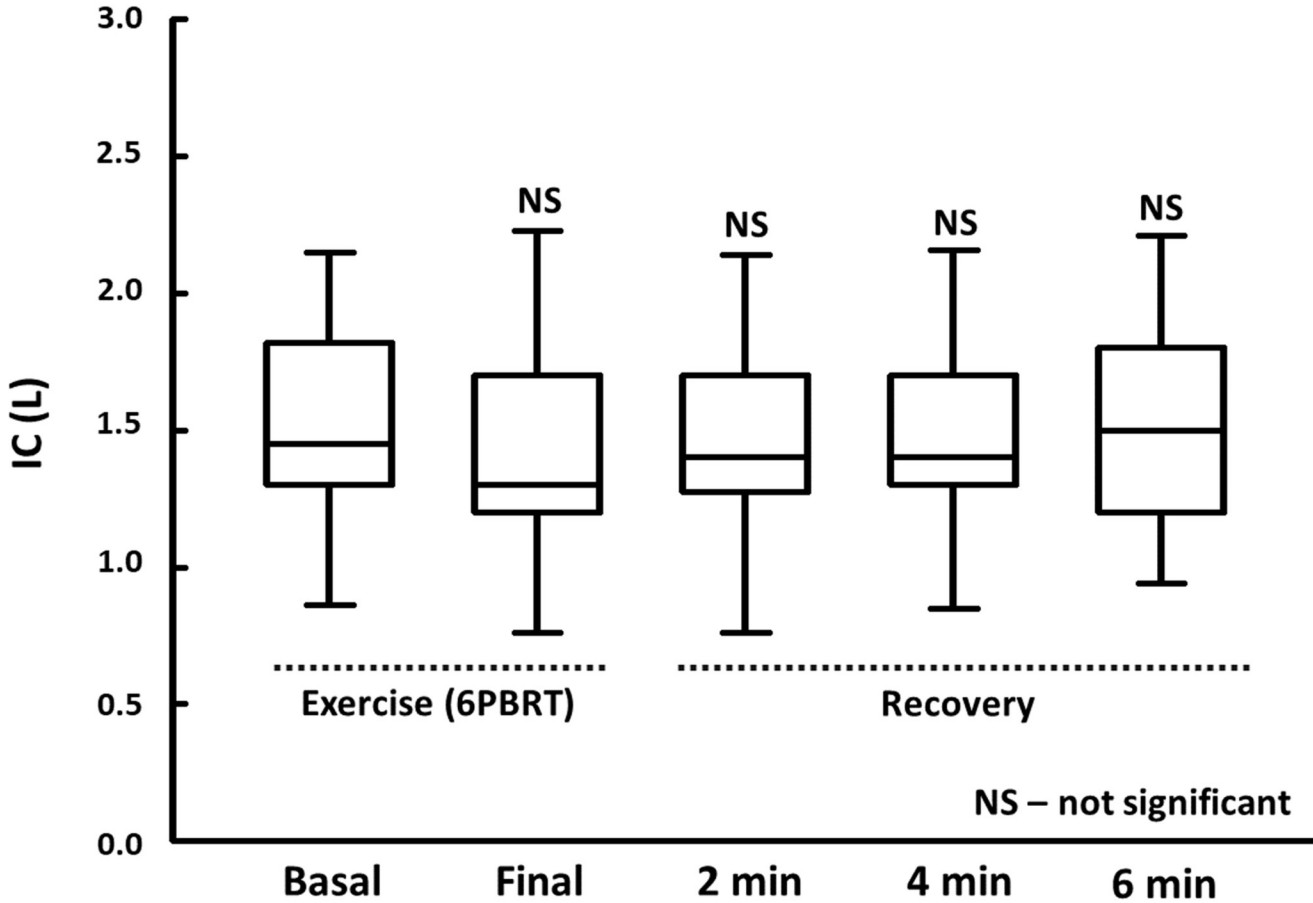

**Fig 3. Serial IC measurements during and after 6-PBRT (repeated measures ANOVA). Abbreviations:** NS, not significant; IC, inspiratory capacity; L, liters; 6-PBRT, six-minute pegboard and ring test; min, minutes.

Finally, for diastolic blood pressure (DBP), we observed a statistically significant difference only immediately after the test (p = 0.04) (**Fig 5**).

## Symptom responses during and after 6-PBRT

We can observe a significant increase in the modified Borg scale for dyspnea and upper limbs fatigue assessment, immediately after the end of 6-PBRT, with dyspnea recovery in 2 minutes of rest and fatigue in 6 minutes of rest (**Fig 6**).

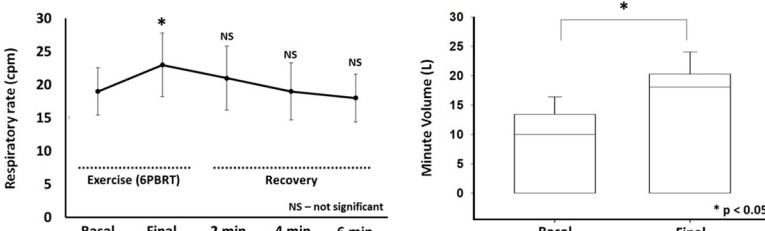

**Fig 4. Respiratory rate and minute volume responses. Abbreviations:** cpm, cycles per minute; 6-PBRT, six-minute pegboard and ring test; NS, not significant; L, liters; min, minutes; * significant difference when compared to basal values.

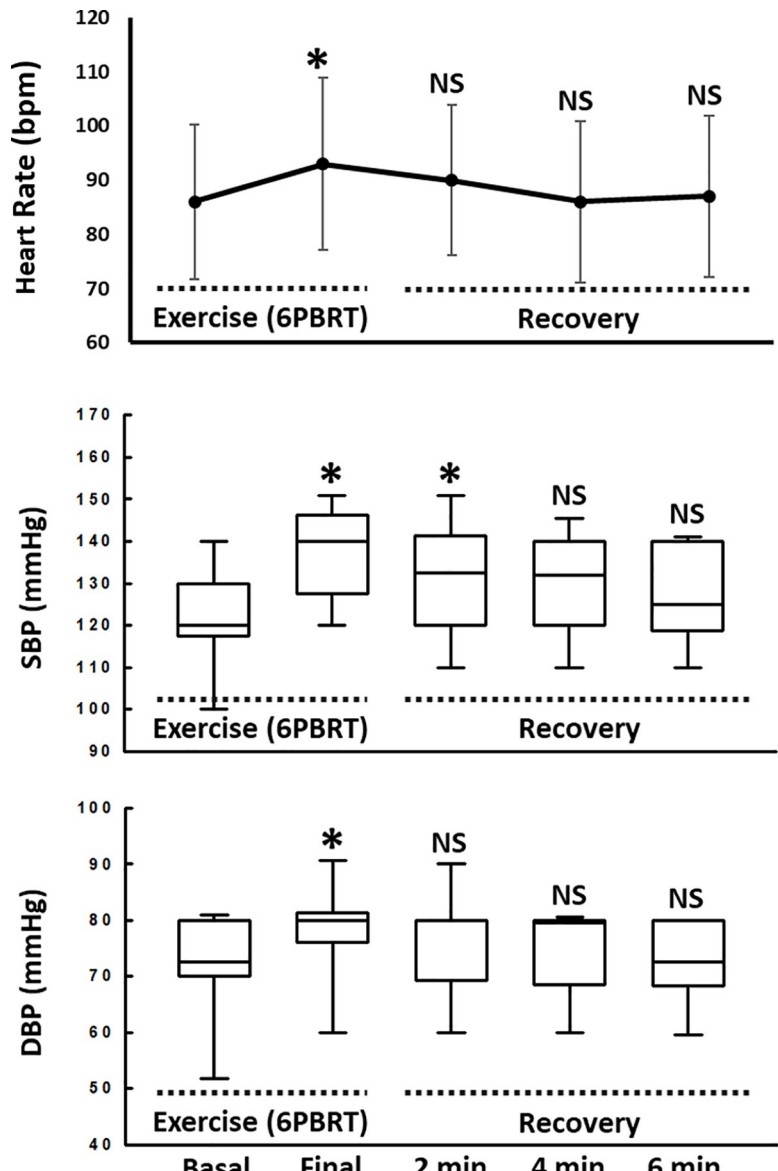

**Fig 5. Heart Rate, systolic and diastolic blood pressure responses. Abbreviations:** bpm, beats per minute; NS, not significant; 6-PBRT, six-minute pegboard and ring test; SBP, systolic blood pressure; DBP, diastolic blood pressure; min, minutes; * significant difference when compared to basal values.

### Feasibility and safety of 6-PBRT in hospitalized patients with AECOPD

All participants (100%) completed the 6-PBRT, which was interrupted by 10 patients (55.5%), but this interruption happened for a few seconds and after the recovery of the symptom that led to the interruption, the patient ran the test again until its conclusion. Dyspnea was the cause of interruption observed in one patient and upper limbs fatigue was the cause of interruption for two patients. Both symptoms were observed as main cause of interruption for the others 7 patients. We observed a maximum interruption time of 20 seconds.

There was no record of adverse events during the 6-PBRT. We evaluated the time of assembly and transportation of the equipment, and it proved to be a device of simple handling, with a mean of $11 \pm 6.72$ minutes between transport and assembly.

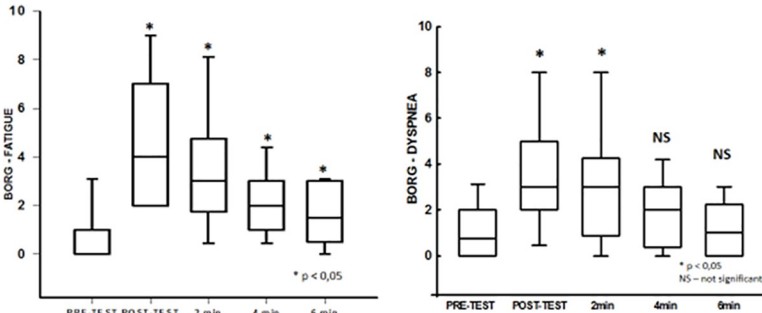

**Fig 6. Upper limbs fatigue and dyspnea responses during and after 6-PBRT. Abbreviations:** 6-PBRT, six-minute pegboard and ring test; min, minutes; NS, not significant; * significant difference when compared to basal values.

## Discussion

The current study allows us to observe the main results: 1) the prevalence of dynamic hyperinflation induced by the 6-minute pegboard and ring test was 50% in hospitalized patients for acute exacerbation of chronic obstruction pulmonary disease (AECOPD); 2) The 6-PBRT induced hemodynamic and ventilatory responses with complete recovery within 6 minutes at the end of the test; and 3) 6-PBRT is safe and feasible to this population.

There are few studies that evidenced dynamic hyperinflation after upper limb exercise and no study has evaluated it after 6-PBRT. In our study the prevalence of DH was 50%, higher than that observed in the first study that described DH during exercises with upper limbs [32]. We believe that this result was observed due to the different modalities of exercise and the fact that 6-PBRT was applied during the exacerbation phase. In a previous study, DH developed during the exercise with cycle ergometer for upper limb was correlated directly to the exercise load. The prevalence of DH in exercise with 50% of the maximum load was 41.1%, while for the loads of 65% and 80%, the prevalence was 66.6% and 79.1%, respectively [22]. In our study, the prevalence was equivalent to that observed by Colluci et al. (2005) during exercise with load between 50–65% of the maximum determined in incremental test [22]. This result suggests that 6-PBRT in patients with AECOPD may correspond to a moderate intensity exercise test.

Our results also demonstrated an increase in respiratory rate after the end of the 6-PBRT, with recovery of baseline values occurring within 2 minutes of rest. Since the increased respiratory rate is one of the main factors for the occurrence of dynamic hyperinflation, we understand that 6-PBRT promotes ventilatory responses in the studied population, but not enough to generate hyperinflation of large magnitude. Our results also agree with previous findings that demonstrated DH and thoracoabdominal asynchrony developed during two techniques (diagonal exercise and cycle ergometer) in patients with stable COPD, and which observed that there was an increase in respiratory rate in both exercise modalities, but which had a statistically greater variation during the exercise with an upper limb cycle ergometer, in which the patients presented DH and more asynchrony [8].

As expected, the minute volume (MV) showed significant increase immediately after the end of the test. It is already known that the mechanisms which promote DH are increased MV, reduced expiratory time, increased elastic work and, finally, decreased respiratory work efficiency. Colluci et al (2010) showed that, regardless of the load stipulated to the exercise, there was an interruption of the exercise when reaching an MV of approximately 39L/min [22]. Castro et al. (2013) found values of 27.9 L/min and 36.7 L/min in two different exercises performed for upper limbs [8]. In another study, Satake et al. (2015) demonstrated a mean

MV of 28.24 ± 9.21 L/min at the peak of exercise, as a possible mechanism causing dyspnea [33]. The mean number found in our study for MV at the end of 6-PBRT was 19.5 ± 3.31 L/min, suggesting that ventilatory mechanisms probably had little impact on the development of dynamic hyperinflation. Despite of this, the intolerance to the test could be confirmed since the patients with AECOPD presented worse performance when compared to the reference values for the normal Brazilian population [31].

Few studies have evaluated the hemodynamic responses during the 6-PBRT. We observed that the heart rate and diastolic blood pressure at the end of the test returned rapidly to their baseline values, requiring only 2 minutes of rest for this recovery. While to reestablish the systolic blood pressure, it took 4 minutes of rest. Only one study evaluated the same parameters immediately after the end of 6-PBRT, finding values similar to ours [15]. However, this study was conducted in an outpatient setting. These results suggest that 6-PBRT does not cause hemodynamic changes outside a safety margin that may limit their execution.

Another important finding of the current study was that the dyspnea was restored faster than fatigue, evaluated by the modified Borg scale. It is already well established in the literature that the sensation of dyspnea and fatigue may be one of the main limiters of unsupported upper limb exercise tolerance in patients with COPD [34, 35] and dyspnea is related to the stimulus levels at the respiratory center and to the magnitude of dynamic hyperinflation [36]. Similar results were demonstrated by Zhan et al. (2006), who observed lower values for a modified BORG scale for the evaluation of dyspnea compared to fatigue in stable patients with COPD [15]. Previous studies have found similar values for fatigue and dyspnea for incremental activity with upper limbs [22] and a linear increase in dyspnea, as well as the decrease in inspiratory capacity during the 6-minute walk test, suggesting that the cause of increased dyspnea assessed by the BORG scale occurred as a consequence of dynamic hyperinflation [33].

Some studies have already shown that the improvement of dyspnea after AECOPD is associated with a reduction in pulmonary dynamic hyperinflation (increase in IC) [37, 38]. Static hyperinflation, which increases even more during exacerbations, is very relevant for understanding the cause of increased dyspnea [39]. In a later study, an assessment of static and dynamic hyperinflation was performed during AECOPD and after a period of stabilization. Patients were evaluated in this trial with a metronome rhythm test aimed to change the IC during tachypnea. This mechanism was sensitive to induce dynamic hyperinflation, but did not find an additional increase beyond the change in static hyperinflation already induced by exacerbation [40].

Regarding the applicability of the test, we observed that the equipment is simple, of fast assembly, and the demand of execution of the 6-PBRT requires only one evaluator. All patients completed the test, with interruption occurring in with 10 patients (55.5% of the sample). All the patients returned and completed the total time of 6 minutes of evaluation, as described in the test methodology. The main reason for interrupting was the fatigue of upper limbs. Felisberto at al. (2018) showed a strong correlation of the performance in 6-PBRT with the increase in fatigue of upper limbs (r = -0.76) [29].

To the best of our knowledge, there are no other studies published in the literature that have evaluated the ventilatory and hemodynamic responses, nor the applicability and safety of 6-PBRT in hospitalized patients with AECOPD. The test demonstrated hemodynamic and ventilatory changes at levels considered safe and with a return to baseline values in a maximum of 6 minutes of rest. The presence of any adverse events that discontinue 6-PBRT definitively has not been demonstrated, showing that 6-PBRT can be considered applicable and safe for this population.

In this context, we reinforce that 6-PBRT is an excellent tool to evaluate the functional capacity of upper limbs and may be included within pulmonary rehabilitation programs

during the acute exacerbation phase of COPD in the hospital setting. In addition, the results of the current study suggest continuing studying the responsiveness to a pulmonary rehabilitation program in the hospital phase, as we know of its ability to evaluate the response to upper limb training in an outpatient setting. However, new studies are necessary to evaluate predictive factors to develop DH.

The main limitations of our study were the number of patients who refused to participate, even though they had similar characteristics to the study population; and the impossibility of evaluating the tidal volume during the test, which would give us more accurate information about the ventilation and mechanisms that cause dynamic hyperinflation. Another important limitation of the study was that we did not evaluated the static hyperinflation. This could have provided ventilatory demand regarding pulmonary function in AECOPD patients and how this phenomenon could influence the magnitude of DH and the frequency of symptoms such as dyspnea during the execution of 6-PBRT.

## Conclusion

The 6-PBRT promotes ventilatory and hemodynamic changes during its execution, requiring a maximum of 6 minutes for its recovery. This test proved to be safe and applicable for hospitalized patients with COPD for acute exacerbation.

## Acknowledgments

The authors acknowledge the Hospital Sírio-Libanês.

## Author Contributions

**Conceptualization:** Cassia Fabiane de Barros, Wellington Pereira Yamaguti.

**Data curation:** Cassia Fabiane de Barros, Rosimeire Marcos Felisberto, Kelly Cristina Albanezi Nucci.

**Formal analysis:** Cassia Fabiane de Barros, Wellington Pereira Yamaguti.

**Investigation:** Cassia Fabiane de Barros, Rosimeire Marcos Felisberto, Kelly Cristina Albanezi Nucci.

**Methodology:** Andre Luis Pereira de Albuquerque, Elaine Paulin, Christina May Moran de Brito, Wellington Pereira Yamaguti.

**Project administration:** Wellington Pereira Yamaguti.

**Resources:** Andre Luis Pereira de Albuquerque, Elaine Paulin, Christina May Moran de Brito.

**Supervision:** Wellington Pereira Yamaguti.

**Validation:** Andre Luis Pereira de Albuquerque, Elaine Paulin, Christina May Moran de Brito.

**Writing – original draft:** Cassia Fabiane de Barros, Wellington Pereira Yamaguti.

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
