## [Decision Letter · Decision Letter 0]

11 Aug 2020

PONE-D-20-20208

Dynamic hyperinflation induced by the 6-minute pegboard and ring test in hospitalized patients with exacerbated COPD

PLOS ONE

Dear Dr. Yamaguti,

Thank you for submitting your manuscript to PLOS ONE. After careful consideration, we feel that it has merit but does not fully meet PLOS ONE’s publication criteria as it currently stands. Therefore, we invite you to submit a revised version of the manuscript that addresses the points raised during the review process. All reviewers raised some significan tconcenrs which are mainly attributed to methodological issues. Furthermore you have to convince us about validity of the current study. 

We look forward to receiving your revised manuscript.

Kind regards,

Stelios Loukides

Academic Editor

PLOS ONE

Journal Requirements:

Reviewers' comments:

Reviewer's Responses to Questions

**Comments to the Author**

1. Is the manuscript technically sound, and do the data support the conclusions?

Reviewer #1: Partly

Reviewer #2: Yes

Reviewer #3: Yes

Reviewer #4: Partly

2. Has the statistical analysis been performed appropriately and rigorously? 

Reviewer #1: Yes

Reviewer #2: Yes

Reviewer #3: Yes

Reviewer #4: Yes

3. Have the authors made all data underlying the findings in their manuscript fully available?

Reviewer #1: Yes

Reviewer #2: No

Reviewer #3: Yes

Reviewer #4: Yes

4. Is the manuscript presented in an intelligible fashion and written in standard English?

Reviewer #1: Yes

Reviewer #2: Yes

Reviewer #3: Yes

Reviewer #4: Yes

5. Review Comments to the Author

Reviewer #1: The manuscript entitled “Dynamic hyperinflation induced by the 6PBRT in hospitalized patients with exacerbated COPD” the authors present the findings that the 6PBRT is safe and applicable for patients hospitalized due to AECOPD.

The data collected in this study contribute to our understanding of this phenomenon and the manuscript could be considered for publication following revisions.

There is a limitation. To overcome this limitation, the authors could do 6PBRT in patients with not only AECOPD and patients with recovered AECOPD. To overcome this limitation, it is necessary that to compare patients with AECOPD and who have recovered from AECOPD.

Please check line 95 and 319, bibliography marks are wrong style.

Reviewer #2: The authors wrote an interesting manuscript regarding dynamic hyperinflation during COPD exacerbations. Very strong is that they used actual physical maneuvers to assess this. The used actual changes from base line to assess the changes which is strong to. Unfortunately the was no information regarding the DH in stable state for the included patients

I do have some comments

Mayor:

Was the trial registered in a public database? What was the predefined primary endpoint for the test? The manuscripts suggests that at first the overall occurrence of DH was the endpoint, which when proved non significant was switched towards cross sectional analysis.

There is no information about the presence of static hyperinflation in the reported patients during the exacerbation, please discuss how this might influence the interpretation of the now presented results

Please report data about the treatment patients were receiving during the exacerbations and the policy regarding the last dose of bronchodilators prior to the measurements.

The PFSDQ-M and CAT were applied to the patients but not reported

Minor

Why was the cutoff of 150 ml chosen rather 100 ml for IC? Please discuss

Please discuss the relation between the presence of static hyperinflation during exacerbation with DH. (For inspiration see e.g DOI: 10.1183/09031936.05.00136304 , DOI: 10.2147/COPD.S154878, DOI: 10.1164/rccm.200504-595OC DOI: 10.1016/S2213-2600(15)00459-2 )

Please discuss the known information and previous trials assessing DH during exacerbations e.g. with other test such as metronome and relate your outcomes to those

Please discuss the lack of data regarding stable state for the interpretation of the results. E.g. what if DH was present in stable state and actually got better or vice versa, and what about static hyperinflation during stable state and the influence on the now presented results

Please extend the baseline data of the patients with more clinical parameters

Please explain the high number of interruptions during the tests

Please explain the high number of dropouts

Reviewer #3: Reason for Reject: A small study and of low impact with a specific task that says something on disease and upper extremity fitness and general fitness therefore, but i do not think that this belongs to a journal with such high impact factor due to rather low priority.

Reviewer #4: Thank you for the opportunity to review this manuscript. I send some suggestions to improve the quality of the study.

The aim of this study was to evaluate physiological responses and Dynamic Hyperinflation induced by the 6-minute pegboard and ring test in hospitalized patients with acute exacerbation of COPD. There is this gap in the literature on the study of the occurrence of dynamic hyperinflation during activities of unsupported upper limbs such as 6PBRT.

The proposal of the study aims to answer this question and its basis is structured by the scientific literature, however I felt a lack of greater theoretical basis justifying its realization. Only 21 studies were used as references for the construction of the text and there are more papers available in the literature that support the ideas pointed out and of great scientific relevance that can contribute.

- Were the values obtained in the pulmonary function test compared to specific reference values for the population?

- The original description of the test should be reported/referenced on line 162.

- Two specific questionnaires related to lung disease were applied (PFSDQ-M and CAT), but the results were not presented.

- 10 patients interrupted the test for the reasons indicated in lines 297-299, however the time of this interruption was not presented, and we know that this time can influence the values obtained. In addition, the information that all patients completed the 6 minutes of the test was only informed in the discussion section. Did any patient complete the 6 minutes of the test at rest?

- The statistical tests that were performed and which variables and moments were considered to calculate must be presented and signaled in the figure 3.

- The number of rings moved during the test was not shown. I believe that this value should be informed to the reader.

- Also, inform the number of rings for patients who presented HD and those who did not. Did patients present HD move a lower number of rings?

Was the number of rings obtained by patients lower than the normal value already established for the population?

- The aim of this study was to evaluate physiological responses and DH induced by 6PBRT in hospitalized patients with acute exacerbation of COPD, however I consider it important to point out if the occurrence of HD was able to provide worse performance in the test and not only if it occurs or not.

- Line 331: “dynamic hyperinflation” = DH.

- Lines 337-339: "The mean number found in our study for MV at the end of 6PBRT was 19.5 ± 3.31 L / min, suggesting that ventilatory mechanisms probably had little impact on the development of dynamic hyperinflation and intolerance to the test". How can the test intolerance be pointed out if we do not know if the patients had lower than expected ring number values?

- Lines 373-376: "The main limitations of our study were the number of patients who refused to participate, even though they had similar characteristics to the study population; and the impossibility of evaluating the tidal volume during the test, which would give us more accurate information about the ventilation and mechanisms that cause dynamic hyperinflation ". There are recent studies that can support the occurrence of greater ventilatory demand impacting patients' performance in tests to evaluate the functional capacity of upper limbs such as 6PBRT.

- The aims must be answered at the conclusion of the study; however, evaluating the safety and applicability of 6PBRT for hospitalized patients with COPD for acute exacerbation was not pointed out as an initial aim of the study.

6. PLOS authors have the option to publish the peer review history of their article (what does this mean?). If published, this will include your full peer review and any attached files.

Reviewer #1: **Yes: **Takeda Kenichi

Reviewer #2: No

Reviewer #3: No

Reviewer #4: No

---

## [Author Response · Author response to Decision Letter 0]

10 Sep 2020

Ms. Ref. No.: PONE-D-20-20208

Title: Dynamic hyperinflation induced by the 6-minute pegboard and ring test in hospitalized patients with exacerbated COPD

Dear Dr. Stelios Loukides,

We really appreciated the contributions made by the review team. We sought to achieve the best possible revision of each of the reviewers’ concerns. We understand that in fact the suggestions were relevant to the improvement of the manuscript and we are available to elucidate further clarification if necessary.

 REVIEWERS’ COMMENTS:

Reviewer #1:

1. The manuscript entitled “Dynamic hyperinflation induced by the 6PBRT in hospitalized patients with exacerbated COPD” the authors present the findings that the 6PBRT is safe and applicable for patients hospitalized due to AECOPD.

The data collected in this study contribute to our understanding of this phenomenon and the manuscript could be considered for publication following revisions.

There is a limitation. To overcome this limitation, the authors could do 6PBRT in patients with not only AECOPD and patients with recovered AECOPD. To overcome this limitation, it is necessary that to compare patients with AECOPD and who have recovered from AECOPD.

Authors: We appreciate this observation. Our study was one of the pioneers investigating the use of 6PBRT as a tool for assessing exercise tolerance in a hospital setting. In this study, we would like to investigate the limitations imposed by the exacerbation in the applicability an

d safety of the test, as well, to understand the prevalence of hyperinflation during the teste. We did not aim to evaluate the patient in the stable or recovered phases. 

2. Please check line 95 and 319, bibliography marks are wrong style.

Authors: We appreciate the reviewer's observation. The change was made in the text.

Reviewer #2:

3. Was the trial registered in a public database? What was the predefined primary endpoint for the test? The manuscripts suggest that at first the overall occurrence of DH was the endpoint, which when proved non significant was switched towards cross sectional analysis.

Authors: We thank the reviewer for the question. Our study had the transversal design aimed to evaluate dynamic hyperinflation with 6-minute pegboard and ring in patients with exacerbated COPD. Primary endpoint was occurrence of DH and the secondary endpoints was hemodynamic and respiratory changes induced by 6PBRT. We emphasize that the functional test chosen in this study was previously described in the literature. This is not a randomized clinical trial. Therefore, the rules of mandatory registration in a public Clinical Trials database are not applicable to our study design. 

4. There is no information about the presence of static hyperinflation in the reported patients during the exacerbation, please discuss how this might influence the interpretation of the now presented results.

Authors: We appreciate the reviewer's observation. Static lung volumes such as residual volume and total lung capacity were not measured. This could have provided ventilatory demand regarding pulmonary function in exacerbated COPD patients. We authors discuss and agree that this is a limitation. We added this information in the “limitation” paragraph of the “Discussion” section.

5. Please report data about the treatment patients were receiving during the exacerbations and the policy regarding the last dose of bronchodilators prior to the measurements.

Authors: We appreciate the reviewer's observation. The authors agree that these notes are relevant and the treatment patients was added and described in the “Materials and methods” section. As a policy regarding the last dose of bronchodilators before measurements, we requested that the short-term bronchodilator should not be performed 2 hours before the tests, ensuring that similar conditions were established for all patients during all tests.

6. The PFSDQ-M and CAT were applied to the patients but not reported.

Authors: We appreciate the reviewer's observation and sorry for this mistake. We added theses complementary information in Table 1 with demographic and spirometric characteristics of the study participants. 

7. Why was the cutoff of 150 ml chosen rather 100 ml for IC? Please discuss.

Authors: We appreciate the reviewer's observation. This cutoff value was used according to the previous study by O’Donnell (2001) and Colucci (2010) 

8. Please discuss the relation between the presence of static hyperinflation during exacerbation with DH. (For inspiration see e.g DOI: 10.1183/09031936.05.00136304 , DOI: 10.2147/COPD.S154878, DOI: 10.1164/rccm.200504-595OC DOI: 10.1016/S2213-2600(15)00459-2)

Please discuss the known information and previous trials assessing DH during exacerbations e.g. with other test such as metronome and relate your outcomes to those.

Authors: We thank the reviewer for this important observation. The primary objective of our study was to verify the occurrence of dynamic hyperinflation during 6PBRT and possible limitations for the execution of the test, such as ventilatory and hemodynamic responses in order to develop a clinical tool for assessing functional capacity during exacerbation. Until data collection there were a few studies carried out with this population at the time of exacerbation during hospitalization. In order, to clarify this relevant point, we added the reference and discussion in the seventh paragraph of the “Discussion” section. We also added this information in the “limitation” paragraph.

9. Please discuss the lack of data regarding stable state for the interpretation of the results. E.g. what if DH was present in stable state and actually got better or vice versa, and what about static hyperinflation during stable state and the influence on the now presented results.

Authors: We appreciate this observation. To ensure the safety of the test we aimed to evaluate the patient during the exacerbation in order to verify the applicability and safety of the test in the hospital setting. In this study, unfortunately, we didn’t evaluate static hyperinflation and we added this information in the “limitation” paragraph.

10. Please extend the baseline data of the patients with more clinical parameters

Authors: We appreciate the reviewer's suggestion. Table 1 with demographic and spirometric characteristics of the study participants has been updated.

11. Please explain the high number of interruptions during the tests.

Authors: We thank the reviewer for the question. All participants (100%) concluded the 6-PBRT, being the same interrupted by 10 patients, but this interruption happened for a few seconds. After the recovery of the symptoms that led to the interruption, the patients ran the test again until its conclusion. Dyspnea was the cause of interruption observed in one patient and upper limbs fatigue was the cause of interruption for two patients. Both symptoms were observed as main cause of interruption for the others 7 patients. We observed a maximum interruption time of 20 seconds. We added these informations in the “Results” section.

12. Please explain the high number of dropouts.

Authors: We are grateful for the question. The explanation of droputs are described in figure 2.

Reviewer #3

13. Reason for Reject: A small study and of low impact with a specific task that says something on disease and upper extremity fitness and general fitness therefore, but I do not think that this belongs to a journal with such high impact factor due to rather low priority.

Authors: The authors would like to request a reconsideration of our manuscript after the comments of reviewers 1, 2 and 4 and the opportunity to respond offered by the editor. All comments have been taken into consideration and we understand that the study has some limitations. We have included all additional data in the objectives, methods and discussion sessions, including the limitations. Thus, we believe we have improved our manuscript. Regarding the sample size we calculated the number of patients that should be included considering the primary objective, according the previous published data of Collucci and O’donell. In addition, we would like to point out that there are no studies evaluating the safety of using 6-PBRT in patients with exacerbated COPD, making it an extremely relevant study for hospital clinical practice.

Reviewer #4

14. Thank you for the opportunity to review this manuscript. I send some suggestions to improve the quality of the study.

Authors: We are grateful for this feedback and suggestions.

15. The aim of this study was to evaluate physiological responses and Dynamic Hyperinflation induced by the 6-minute pegboard and ring test in hospitalized patients with acute exacerbation of COPD. There is this gap in the literature on the study of the occurrence of dynamic hyperinflation during activities of unsupported upper limbs such as 6PBRT.

The proposal of the study aims to answer this question and its basis is structured by the scientific literature; however I felt a lack of greater theoretical basis justifying its realization. Only 21 studies were used as references for the construction of the text and there are more papers available in the literature that support the ideas pointed out and of great scientific relevance that can contribute.

Authors: We thank the reviewer for this important observation. For better understanding, we rewrote the third and fourth paragraph of the section "Introduction" and we complement it with references from important studies on the subject.

16. Were the values obtained in the pulmonary function test compared to specific reference values for the population?

Authors: We thank the reviewer for the question. In the present study we follow the reference according to American Thoracic Society (ATS)/ European Respiratory Society to perform spirometry. For better understanding, we rewrote the sentence about assessment in the “Materials and methods” section.

17. The original description of the test should be reported/referenced on line 162.

Authors: We appreciate the reviewer's suggestion. This information was included in the “Materials and methods” section.

18. Two specific questionnaires related to lung disease were applied (PFSDQ-M and CAT), but the results were not presented.

Authors: We appreciate the reviewer's observation. Table 1 with demographic and spirometric characteristics of the study participants has been updated with this information.

19. 10 patients interrupted the test for the reasons indicated in lines 297-299, however the time of this interruption was not presented, and we know that this time can influence the values obtained. In addition, the information that all patients completed the 6 minutes of the test was only informed in the discussion section. Did any patient complete the 6 minutes of the test at rest?

Authors: We thank the reviewer for the question. All participants (100%) concluded the 6-PBRT and no patient completed the test at rest. We observed interruptions during the test by 10 patients, and these interruptions happened for a few seconds. The maximum interruption time observed in a test was 20 seconds. For better understanding, we rewrote this sentence in the “Results” section.

20. The statistical tests that were performed and which variables and moments were considered to calculate must be presented and signaled in the figure 3.

Authors: We appreciate this observation. We added this information in the legend in Figure 3.

21. The number of rings moved during the test was not shown. I believe that this value should be informed to the reader.

Authors: We appreciate the reviewer's observation. We added the data in the “Results” section.

22. Also, inform the number of rings for patients who presented HD and those who did not. Did patients present HD move a lower number of rings?

Authors: We thank the reviewer for the question. Patients who hyperinflated in the test presented worse performance when compared to those who did not hyperinflate (230.22 ± 68.56 vs 259.55 ± 57.49, respectively). However, this result was not significant (P = 0,34). We added this information in the “Results” section.

23. Was the number of rings obtained by patients lower than the normal value already established for the population?

Authors: We thank the reviewer for the question. The performance in 6-PBRT was worse in patients with exacerbated COPD when compared to the predicted values for normal population. We added this information in the “Results” section.

24. The aim of this study was to evaluate physiological responses and DH induced by 6PBRT in hospitalized patients with acute exacerbation of COPD, however I consider it important to point out if the occurrence of HD was able to provide worse performance in the test and not only if it occurs or not.

Authors: We really appreciate the reviewer's suggestion. These results were added to the manuscript.

25. Line 331: “dynamic hyperinflation” = DH

Authors: We appreciate the reviewer's observation. The change was made in the manuscript.

26. Lines 337-339: "The mean number found in our study for MV at the end of 6PBRT was 19.5 ± 3.31 L / min, suggesting that ventilatory mechanisms probably had little impact on the development of dynamic hyperinflation and intolerance to the test". How can the test intolerance be pointed out if we do not know if the patients had lower than expected ring number values?

Authors: We really appreciate the reviewer's observation. We added this information in the “Results” and “Discussion” sections.

27. Lines 373-376: "The main limitations of our study were the number of patients who refused to participate, even though they had similar characteristics to the study population; and the impossibility of evaluating the tidal volume during the test, which would give us more accurate information about the ventilation and mechanisms that cause dynamic hyperinflation ". There are recent studies that can support the occurrence of greater ventilatory demand impacting patients' performance in tests to evaluate the functional capacity of upper limbs such as 6PBRT.

Authors: We appreciate this observation. We observed and described studies that showed this increase in ventilatory demand, but so far there are no studies that demonstrate this phenomenon with the patient in hospital for AECOPD.

28. The aims must be answered at the conclusion of the study; however, evaluating the safety and applicability of 6PBRT for hospitalized patients with COPD for acute exacerbation was not pointed out as an initial aim of the study.

Authors: We appreciate the reviewer's observation. We note that this objective was not reported in the summary and introduction section. This information was added in the manuscript.

---

## [Decision Letter · Decision Letter 1]

19 Oct 2020

Dynamic hyperinflation induced by the 6-minute pegboard and ring test in hospitalized patients with exacerbated COPD

PONE-D-20-20208R1

Dear Dr. Yamaguti,

We’re pleased to inform you that your manuscript has been judged scientifically suitable for publication and will be formally accepted for publication once it meets all outstanding technical requirements.

Kind regards,

Stelios Loukides

Academic Editor

PLOS ONE

Additional Editor Comments (optional):

Reviewers' comments:

Reviewer's Responses to Questions

**Comments to the Author**

1. If the authors have adequately addressed your comments raised in a previous round of review and you feel that this manuscript is now acceptable for publication, you may indicate that here to bypass the “Comments to the Author” section, enter your conflict of interest statement in the “Confidential to Editor” section, and submit your "Accept" recommendation.

Reviewer #1: All comments have been addressed

2. Is the manuscript technically sound, and do the data support the conclusions?

Reviewer #1: Yes

3. Has the statistical analysis been performed appropriately and rigorously? 

Reviewer #1: Yes

4. Have the authors made all data underlying the findings in their manuscript fully available?

Reviewer #1: Yes

5. Is the manuscript presented in an intelligible fashion and written in standard English?

Reviewer #1: Yes

6. Review Comments to the Author

Reviewer #1: The manuscript entitled “Dynamic hyperinflation induced by the 6PBRT in hospitalized patients with exacerbated COPD” the authors present the findings that the 6PBRT is safe and applicable for patients hospitalized due to AECOPD.

The data collected in this study contribute to our understanding of this phenomenon and the manuscript could be considered for publication following revisions.

The authors report 6PBRT in patients with only on AECOPD. Please report data about the recovered pulmonary function test after hospitalization patients with AECOPD. And It is necessary that 6PBRT data in patients with recovered AECOPD. It will help us to understand how severe those patients with AECOPD and COPD.

7. PLOS authors have the option to publish the peer review history of their article (what does this mean?). If published, this will include your full peer review and any attached files.

Reviewer #1: **Yes: **Kenichi Takeda

---

## [Editor Report · Acceptance letter]

23 Oct 2020

PONE-D-20-20208R1 

Dynamic hyperinflation induced by the 6-minute pegboard and ring test in hospitalized patients with exacerbated COPD 

Dear Dr. Yamaguti:

I'm pleased to inform you that your manuscript has been deemed suitable for publication in PLOS ONE. Congratulations! Your manuscript is now with our production department. 

Kind regards, 

on behalf of

Dr Stelios Loukides 

Academic Editor

PLOS ONE